# Addressing the Insufficiencies of the Traditional Development Aid Model by Utilizing the One Belt, One Road Initiative to Sustain Development in Afghanistan

**Yanzhe Zhang \*, Xiao Yu and Huizhi Zhang \***

Northeast Asian Studies College, Jilin University, Changchun 130012, China; yux@jlu.edu.cn
\* Correspondence: yanzhe_zhang@jlu.edu.cn (Y.Z.); huizhi@jlu.edu.cn (H.Z.)

**Abstract:** This paper investigated the use of the One Belt One Road initiative (BRI) as a policy model that might address the insufficiencies of the traditional development aid model in reconstructing and developing Afghanistan. Afghanistan has emerged as one of the world's most fragile and conflict-affected countries, and it has gained the attention of both academic and political communities since the early 2000s. The materials for this article are based partly on a thorough analysis of the available documentation. The authors also conducted interviews with high-ranking political elites and policy officials in the Afghan government and international organizations. The study employed a purposive sampling method to identify people with firsthand information on how to sustain economic development in Afghanistan. This paper provides new insights by comparing the traditional development aid model and the BRI in terms of social economy, local security and regional economic development. The aim of this research is to evaluate whether the BRI can remedy the insufficiencies of the traditional development aid model in order to sustain development in Afghanistan. The findings provide a better understanding of the BRI in promoting the internal dynamism required to develop the regional economy, and fill a gap in the literature with regard to the applied and theoretical economic growth models for stabilizing and sustaining the development of fragile and conflict-affected states.

**Keywords:** traditional development aid model; BRI; sustainable development; conflict-affected states; Afghanistan

## 1. Introduction

The One Belt One Road initiative (BRI) was first proposed by Chinese president Xi Jinping in 2013. The idea was originally developed from the ancient concept of "The Silk Road". The BRI included construction of a "Silk Road economic belt" and the "21st century maritime Silk Road". The BRI aimed to build a platform for economic cooperation by creating political trust, economic integration, and cultural compatibility. More than 3000 constructive projects have been put into operation since the BRI was initiated. For example, the Chinese government and enterprises have made foreign direct investments in 49 countries on the Road. The total amount of these international outsourcing contracts amounts to USD 17.83 billion. The primary aim of the BRI is to set up cooperation and promote development among the countries on the Road. This is also in line with the trends in globalization that aim to build a worldwide community sharing common interests and goals. Furthermore, it is important that the Chinese government and enterprises understand that the characteristics of the economic, political, and social environments of the on-Road countries are very unique and could act as

the key factors that influence the outcomes of the BRI. The Chinese government must carefully identify these factors in order to make the right decisions, especially in failed states.

Since the late 1970s, Afghanistan has been recognized as a quintessential failed state [1,2]. Following the events of 11 September 2001, international communities have realized that fragile and conflict-affected states such as Afghanistan can pose threats to regional security and global development. Although billions of dollars flow into Afghanistan in the form of foreign aid, the Afghan people still suffer from substandard living standards, a high unemployment rate, and the potential for the eruption of new conflicts. From an economic perspective, the use of the traditional development aid model to sustain the development of Afghanistan has been a failure. Both internal and external circumstances have damaged the stability and sustainable development of Afghanistan. The question of how best to rebuild the economy and national security in fragile and conflict-affected states has been a focus of academic and political communities internationally.

The theoretical framework of this paper is the fragile state theory. The fragile state theory adopted here, states that countries such as Afghanistan not only fail on the human indicator index, but also require an external economic approach to facilitate sustainable development. The theory provides the framework for assessing the empirical results of this study through the qualitative elite interviews. The results of the interviews demonstrated that the use of the traditional development aid model in Afghanistan has failed; they also helped to establish that the BRI is integral to establishing a new development approach and facilitating internal dynamism to develop the regional economy. The objective of this paper is to explore the insufficiencies of the traditional development aid model that have led to its failure to achieve social stability and sustain the economic development of Afghanistan. It further aims to clarify that the BRI can be used to supplement the traditional development aid model. Moreover, it also aims to verify that the BRI is more effective than the traditional development aid model in facilitating internal dynamism for promoting economic development.

The findings of this study support the novel policy approach that is expressed in the concept of the BRI. The initiative situates Afghanistan as a regional trade and transit hub in the Belt and on the Road. The projects constructed through the BRI have significantly increased Afghanistan's per capita gross domestic product, government revenues, and employment in Afghanistan. Moreover, the study found that Afghanistan must reduce its dependence on traditional development aid and facilitate internal economic growth to promote sustainable development. Overall, the novelty of this paper is that it presents the first comprehensive study of the BRI as a policy approach that facilitates the sustainability of conflict-affected states such as Afghanistan.

The remainder of this paper is organized as follows. Section 2 covers the research area, methodology and data collection; Section 3 provides a literature review of prior studies; Section 4 provides an analysis of Afghan economic history and the reason why the traditional development aid model has failed to promote the Afghan economy effectively; Section 5 explains how the BRI can supplement or remedy the traditional development aid model in sustaining the development in Afghanistan; and the conclusions are drawn in the final section.

## 2. Materials and Methods

Most previous studies have been conducted using the traditional development aid model as an approach that promotes the economic development. However, this approach lacks an in-depth understating of the historical issues and the characteristics of Afghanistan that limit the economic development of Afghanistan. In order to understand the situation and establish an economic approach to promote the sustainable development in Afghanistan, a methodology should be constructed to explore the factors that underlie this study. When investigating the deficiencies in the traditional development aid model and the potential solution to address this problem, the selection of an appropriate research design is critical. It is important to reiterate the research questions that this paper seeks to test and analyze.

Fragile and conflict-affected states tend to look for an approach that facilitates internal dynamism to develop their economy and society while foreign aid is reduced. Hence, this research addressed four key questions. First, what is the situation and characteristics of the Afghanistan's economy? Second, what are the insufficiencies that lead to the failure of the traditional development aid model in Afghanistan? Third, how does the BRI remedy the deficiencies in the traditional development aid model and sustain social and economic development in Afghanistan? And finally, what is the core difference between the traditional development aid model and the BRI in sustaining the economy of Afghanistan?

Based on the significant gap in the current research and practice, this paper aimed to assess the use of the BRI to address the 'blind-spots' in the traditional development aid model in order to stabilize and sustain the development in Afghanistan. Three of core assumptions implicit in the traditional development aid model literature were questioned. First, the traditional development aid model is limited by the international contractors and donors who provide the foreign aid. Second, the internal dynamism of economic development has not been emphasized in the traditional development aid model. Third, the traditional development aid model ignores social problems such as poverty and unemployment, and the sustainability of the economy, which the BRI concentrates on.

This paper, therefore, applied a mixed method for collecting data. The authors used relevant data concerning the national security, economy, politics and social issues in Afghanistan, from international organizations, non-governmental organizations (NGOs), governments including the United Nations (UN), the United States Geological Survey (USGS), the Asia Foundation's Annual Poll, the Word Bank (WB), etc. The study drew extensively on Afghan government documents such as the Afghanistan National Development Strategy (ANDS). The author conducted personal interviews with professionals and the high-ranking politicians who are or had worked as the governors in the international or Afghan institutes. The questions were very relative to the BRI and ANDS. The duration of the interviews spanned around 1–2 h. The author was allowed to take notes in English.

## 3. Literature Review

Since the millennium, the security and sustainability issues experienced by conflict-affected states have been recognized by international communities [3]. Fragile states are usually defined as existing when "the state fails to protect its citizens from violence, to provide services to all citizens, and to be recognized as legitimate by citizens" [4]. However, because there is no agreed-upon definition within the international community of what constitutes a fragile state, each major donor has defined the term to fit its own agenda [5]. The United Nations (UN) generally uses three criteria to define fragile states; there is low income, the Human Assets Index and the Economic Vulnerability Index [6].

In 2011, the former United Nations Secretary General Ban Ki-Moon emphasized that "no conflict-affected country has achieved even one of the Millennium Development Goals" [7]. As can be seen in Table 1, Afghanistan is one of the countries that most demonstrate fragility, post-war conflict, and extreme poverty [8]. Afghanistan is "significantly more prone to large-scale violence and civil conflict than other low-income countries" [9]. It qualifies as a fragile state with a score of 2.6 out of 6.0 on the multi-donor country policy, which is an institutional assessment rating based on economic management, structural policies, social inclusion and equity, and public sector management and institutions. Afghanistan ranked in the top 10 countries out of 177 in the failed state index (FSI) in 2002 [10]. In 2009, Afghanistan was one of 48 countries on the least developed countries index, having been first listed in 1971 [11,12].

**Table 1.** World Bank list of fragile states, FY 2007.

| Level | |
|---|---|
| Severe | Afghanistan, Central African Rep., Comoros, Côte d'Ivoire, Liberia, Myanmar, Somalia, Togo, Zimbabwe |
| Core | Angola, Burundi, Chad, Dem. Rep. of the Congo, Congo, Eritrea, Guinea, Guinea Bissau, Haiti, Kosovo (territory), Lao P.D.R., Solomon Islands, Sudan, Timor-Leste, Tonga, Uzbekistan |
| Marginal | Cambodia, Djibouti, Gambia, Mauritania, Nigeria, Papua New Guinea, São Tomé and Principe, Sierra Leone, Vanuatu |

*Source*: IDA, World Bank 2007, "IDA Operational Approaches and Financing in Fragile States", p. 25.

In order to promote a stable and sustainable economy, Afghanistan has been heavily dependent on international aid to secure both societal and economic development. Since 2002, Afghanistan has received "approximately USD 100 billion" from the United States in non-military aid [13]; however, "the core of the aid architecture—the set of rules governing aid flows—has changed little over the past few decades . . . and the developments of the last decade have radically reshaped the aid environment and the traditional arguments no longer hold" [14]. The traditional development aid model has been criticized as having several issues such as simplicity [15], unsustainability [16], and dependency [17]. Ghani and Lockhart stated, "[T]he expenditure of tens of billions of dollars over a half a century has resulted only in disenchantment and mutual recrimination without many significant breakthroughs in wealth creation" [15]. As the United Nations High Level Forum on Aid Effectiveness concluded:

"The current ways of working in fragile states needs serious improvement. Despite the significant investment and the commitments of the Paris Declaration on Aid Effectiveness (2005) and the Accra Agenda for Action (2008), results and value for money have been modest [7]."

In summary, the traditional development aid model is now out of synch with the situation of the contemporary world [15].

So far, no scholars have made any academic breakthroughs in promoting the sustainable development of failed states. In order to address this critical gap in the existing literature, this paper combines the conceptual propositions underpinning conflicts-affected states, the internal dynamism of economic growth, and the BRI [18]. The understanding of conflict-affected states in this paper is heavily influenced by Evans and Barakat [19], who stated that the traditional development aid model can be an obstacle for the development of conflict-affected states. They emphasized that the outcomes of the traditional development aid model are different when compared to the initial primary goals. Evans and Barakat further argued that the solutions must be localized to solve indigenous problems [19].

The term "internal dynamism of economic growth" refers to the endogenous power of the economy to produce goods and services as the determinant to promote sustained economic growth [20]. It is an essential part of modern development theory. Todaro highlighted three issues that contribute to the sustained economic growth of any country: capital, labor, and technological progress [21].

BRI is expected to boost endogenous economic growth in failed countries. It is a platform that facilitates trade, investment, alignment, and coordination of development strategies in the countries that are on the Silk Road. It aims to tap the potential of the market, promote investment, consumption, and employment, and bring peace and prosperity into the countries alongside the Road [22]. Since the early 2013, Chinese enterprises have built 82 overseas economic and trade cooperation zones in countries along the Belt and Road, with a total investment of USD 28.9 billion, and they have created 244,000 local jobs [23].

The BRI is more advanced than the traditional development aid model. Wu argued that the BRI focuses on building production capacity cooperation to enhance the countries' internal dynamism of economic growth by improving production ability and innovation ability, and promoting countries more deeply into the European, East Asian, and global supply and industrial chains to improve the status of the global value chain of these countries [24].

In summary, the BRI facilitates the internal dynamism of economic growth in conflict-affected states such as Afghanistan. The initiative was designed to make improvements in transportation and infrastructure that will increase trade and the job opportunities. It does not only facilitate GDP growth, but also improves the living standard in Afghanistan. Given this evidence, this paper clarifies the ways in which the insufficiencies of the traditional development aid model are improved by utilizing the BRI approach. The results also provide theoretical support that social stability and economic sustainability might be achieved through BRI in conflict-affected states.

## 4. Historical Afghan Economic Development

### 4.1. The Afghan Economy before 1978

Agriculture is the dominant industry in Afghanistan. Before 1978, more than 90% of Afghans, estimated as 22 million people, were living off harvesting crops and animal production. They traded the surplus for the necessities of life. Their per-capita incomes were similar to most other developing countries, and rural people were almost self-sufficient. Less than 10% of the population lived in urban spaces [25]. In general, the industrial structure of the Afghan economy was unbalanced. This caused the segregation of the central government from a high proportion of citizens and local communities. It also led to three major problems. Firstly, it created a cultural gap between rural and urban people in the administrative sense; secondly, this phenomenon weakened the authority, and then, the function of the central government in the mind of rural citizens; and finally, this would lead to famine in the years when hazards occurred.

In the first half of the 20th century, much of the Afghan revenue was composed of foreign aid received mainly from Germany, Japan, and Italy. Since the beginning of the 1950s, the former Soviet Union and United States have been the major donors to Afghanistan. In 1973, the amount of aid flow generated from the former Soviet Union was USD 428 million [26], which was almost two-thirds of the government revenue. The aid flow from the former Soviet Union increased to USD 600 million in 1975. The total amount of aid from the United States was USD 470 million in 1977 [27] (Dr. Anwar Ahady: a politician in Afghanistan who formerly served as the Afghan Minister of Commerce and Industry). As a sovereign state, the aid flow was managed by the Afghan government, and at that time, foreign contractors and donors acted only as funders and collaborators.

Most of the foreign aid was used to construct basic infrastructure for promoting economic development, and included the building of transportation networks, irrigation systems, electricity grids, etc. These facilities improved the capacity for production in both the agricultural sector and industry. The Afghan government generated revenue through taxes on agricultural exports. Consequently, the income of private and state-owned enterprises (SOEs) increased, which led to sustainable development in Afghanistan [28–30] (Dr. Amin Farhang: a professor of the Afghan economy). As can be seen in Figure 1, the economic status of Afghanistan was stable and sustainable. Ostensibly, the gross domestic product (GDP) was strong, but real economic growth was low.

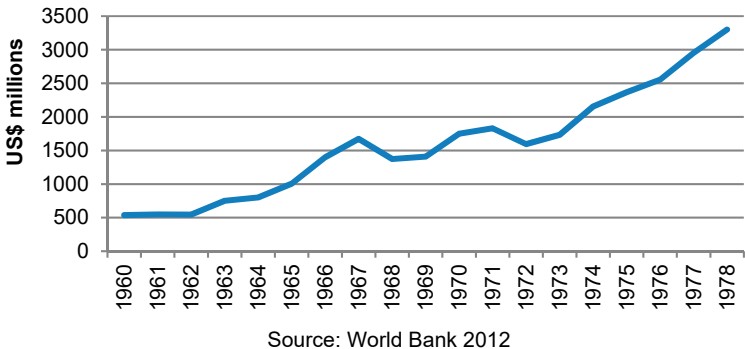

Source: World Bank 2012

**Figure 1.** Nominal GDP, 1960–1978.

### 4.2. The Afghan Economy between 1978–2001

After King Zahir Shah was overthrown in 1973 and Mohammed Daud Khan was killed, the 230 year-long Durrani dynasty ended in 1978 [31]. The communists of the former Soviet Union started to govern Afghanistan and signed a contract that offered USD 250 million in military aid to the Afghan government in July1978. The communists initiated a series of reforms including abolishing the current land rental system and the labor employment system, etc. Government officials replaced those systems with rural cooperatives [25]. Production incentives for rural people were reduced, which consequently led to strong opposition. In order to avoid the downfall of the Communist regime, the Soviet Union invaded Afghanistan in December 1979 [28].

The invasion and war disrupted economic activities, and the foundations of economic growth were destroyed. Due to the air-attacks, the infrastructure was severely damaged and millions of people became homeless. For example, 75% of farming and irrigation areas were destroyed between 1979 and 1989 [25]. Mandatory military conscription by the Afghan government forced a large number of laborers to flee, thus changing the composition of the labor force.

In this situation, Afghanistan's economy became much more dependent on the former Soviet Union. However, the former Soviet Union focused its aid projects on the energy, mining and transport sectors for its own benefit. The products of the aid projects were traded to the former Soviet Union at low prices, while the foreign exchange reserve of Afghanistan decreased significantly. By the early 1980s, Afghanistan owed USD 3 billion to the former Soviet Union, and the revenue from exports was largely used to repay the debt. The former Soviet Union withdrew its army from Afghanistan in 1989 and the Communist regime resigned from power in 1992 [28]. However, further chaos had a considerable influence until 1996.

In 1996, the Taliban took over Afghanistan; however, the country's economic status had not recovered. Agricultural activities had all been abandoned, which led to famine, economic hardship, and violence in the late 1990s. The populace relied heavily on foreign food aid delivered by humanitarian agencies for basic nutrition. Afghanistan has been absent from global development since the late 1970s. The government had barely functioned, and the development of society and economy had been sluggish since 1979.

### 4.3. The Afghan Economy from 2002–2012

After the 9/11 attack, the immediate actions of the U.S. military and other international forces overthrew the Taliban regime. The period of violence and warlords was ostensibly terminated, which led to peace and the preconditions for internal economic growth. Three million Afghans who lived abroad returned to Afghanistan with their assets after 2001, and around 600,000 exiled people returned home and repossessed their land [32]. The international communities provided billions of dollars in foreign aid to assist Afghanistan in rebuilding its society. Since 2001, the Afghan government has received USD 78 billion in foreign aid from international contractors and donors [33]. As Figure 2 shows, the economic status of Afghanistan developed strongly during this period. In 2003, Afghanistan's GDP was only USD 4.1 billion, but it reached USD 12.5 billion in 2009 [34].

Although the GDP has increased, Afghanistan's economic status lacks stability and sustainability. The growth in GDP and economic development were actually over-reliant on the amount of received aid. For example, the overall amount of foreign aid was 97% of Afghanistan's GDP in 2011 [35,36]. This means that the traditional development aid model had been a major factor in facilitating Afghan economic activities. This limited the capability of the Afghan government in explore its internal dynamism of economic growth and to manage the risks when the amount of foreign aid was reduced. The Afghan government lost its authority to control and manage projects that were sponsored by international contractors and donors. In short, the aid flow led to negative influences in Afghanistan, such as Dutch disease and economic collapses and conformed to the traditional development aid model rather than being Afghan-centric.



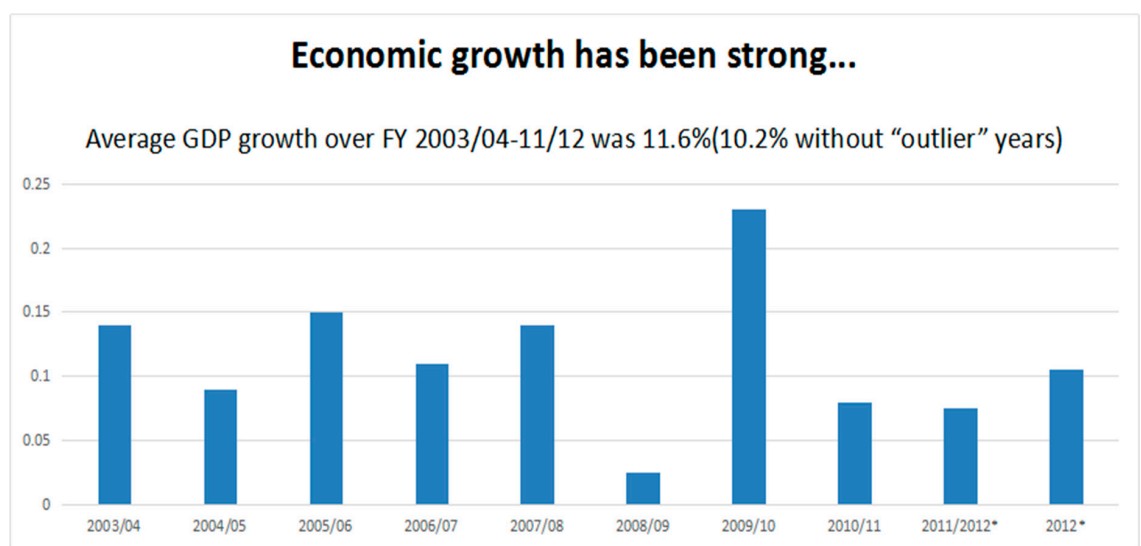

**Figure 2.** Economic growth. *Source*: Farhadi 2013, p. 9.

## 4.4. The Afghan Economy after 2013 and Beyond

Contemporary studies of Afghanistan's economy and society have mainly focused on the future situation when foreign aid withdraws. The traditional development aid model has supported economic development to maintain a high growth rate that averaged 9% from 2003 to 2012. In the financial year 2012–2013, the GDP of Afghanistan was USD 19.8 billion and the per-capita income was around USD 660 [37]. In 2013, the amount of foreign aid was significantly reduced, and this drove economic growth to a stop. Afghanistan's economic growth rate was just 2% in 2013, dropped down to 1.3% in 2014, and only rose to 1.9% in 2015 [38]. In 2016, the GDP declined by 50%. The economic growth rate was already lower than the population growth rate. The unemployment rate was at 40% in 2015, with a year-to-year growth of more than 15%. Furthermore, 36% of the population was living below the poverty line [39].

With plummeting levels of foreign aid, the collapse of the Afghan economy, which is dependent on foreign aid and lacks internal dynamism, is inevitable. This phenomenon has negatively affected the livelihood of ordinary people in Afghanistan. According to the Asia Foundation's Annual Poll, 29.7% of the Afghans thought their living conditions were bad in 2015 [40]. This is the highest proportion on record in Afghan history. Serious corruption problems also dragged down the country's economic development and had a negative impact on people's confidence in national reconstruction.

Overall, the past few decades have seen a lack of economic development in Afghanistan. In the second half of the 20th century, Afghanistan was engulfed in chaos caused by regime changes, invasion, and the actions of warlords. In the aftermath of 9/11, Afghanistan was depicted as a country of persistent conflicts. The frequent suicide bombings and assassinations, and mounting civilian and military casualties produced the image of Afghanistan as a savage rejecting a path to development.

Conversely, the people of Afghanistan desperately hope for a chance of living in peace in order to satisfy the most basic needs of life for their future descendants. They are tired of war and factionalism and are intent on choosing freedom and a path to economic and social development. International communities have primarily sponsored approaches and models that were designed to reduce poverty by improving the functions of government, political legitimacies, human capital, and material resources. The foundation of a development aid model in Afghanistan lies in the key issues of this strategy: providing the conditions for Afghan development and providing the basic financial support to deliver projects that matter to them. However, the efficiency of the traditional development aid model in Afghanistan is questionable because of its simplification.

## 4.5. Failure of the Traditional Development Aid Model in Afghanistan

The main aim of this study was to explore the 'blind-points' of the traditional development aid model and the financial gaps that might threaten development, national security, and other core governmental functions in Afghanistan. In 2000, the public financial management system was eliminated. Although the capacity of the Afghan government and the public financial management system has been upgraded significantly since 2002, the authority for managing and controlling aid funds has not been handed over to Afghanistan.

In 2009, Afghanistan fell to 181 out of 182 countries in the ranking of the global Human Development Report [41]. Due to endless conflicts and prolonged isolation, Afghan society had been fragmented and destroyed. The social capital and public institutions only had limited capacity to address development issues and the new government barely functioned. In managing these problems, international communities have generally focused on using the traditional development aid model to assist with the economic development of failed states such as Afghanistan. With a population of 36.4 million people, 42.3% of them were under 15 years old in 2015 [42]. According to its 2017 annual report, per-capita income was only USD 800 and the unemployment rate stood at 68% [43]. The huge number of youth and high unemployment rate poses threats to the Afghanistan's national security. Thus, security issues have become the overriding priority. A consensus that the traditional development aid model has positively affected few and poses a barrier to Afghanistan's reconstruction has been recognized not only amongst academia and policy makers, but also within the international community. This has several serious implications.

The first implication is that the traditional development aid model has weakened the legitimacy and authority of the Afghan government. Rebuilding government legitimacy is a top priority in fragile and conflict-affected states, but the way in which aid is delivered into Afghanistan has undermined it [44]. This has caused a heavy, enduring dependency on foreign aid and even more unstable circumstances in Afghanistan. Failure to establish an approach that promotes employment, government revenues, and incomes sufficient for Afghans to support their country's growing population greatly increases the likelihood of intense insurgency and internal conflict. Until the economy is divorced from international aid, there is an increased risk that the current conflicts and the turbulent social situation will spill over into civil war.

Second, the traditional development aid model does not help to develop the capacity of the Afghan government. The model should incorporate the administrative resources of Afghanistan in accordance with its national development strategies. However, the Afghan government has limits in accessing most international aid projects. The international contractors and donors who provide the aid are barely involved with and even bypass the government. Hence, it is difficult to imagine how the capacity of the government can be improved under the traditional development aid model. The Afghan government has no authority to make decisions for using the funds for the aid projects, which account for a high proportion of the Afghan GDP [16]. Therefore, the government cannot promise the stabilization of economic growth.

The third implication is that the traditional development aid model poses risks in society once that aid is reduced. In Afghanistan, aid funds provide the majority of the governmental budget for maintenance, supply, personnel, spending for equipment, training its forces, and other costs. For example, the cost of maintaining the Andkhoy-Leman Road is at least USD 200 million annually [45]. In 2011, this was almost 8% of the government's core budget. Although the model provides limited jobs for the Afghan people and stimulates Afghan economic growth, it is hard for the Afghan government to stabilize its economic status and sustain social development in the long-term.

Fourth, the traditional development aid model forces a "one size fits all" model on Afghanistan, thus ignoring the complexities of Afghanistan. The primary aim of the model is to strengthen and reform the public sector institutions. It is designed to support and assist a weak government that has basic administrative standards and functions. In order to obtain foreign aid, the Afghan

government has prioritized meeting the demands and conditions of international contractors and donors, rather than satisfying the needs of the people.

Finally, the traditional development aid model has failed to stabilize the society in Afghanistan. Extreme poverty and joblessness are the most serious problems and have caused conflicts in Afghanistan since the late 1970s [39]. At nearly half of the total populace, few young people can undertake higher education in Afghanistan. Uneducated and unemployed youth are filled with dissatisfaction with the government and are a danger to society, thus they pose highly potential risks to national security and sustainable development in Afghanistan. Hence, the traditional development aid model does not work to solve poverty and the problem of unemployment.

In short, international contractors and donors have crippled the functioning of Afghan institutions. The traditional development aid model has several structural shortcomings that have been outlined above. Afghanistan has to make an inevitable decision: either rely on unsustainable foreign aid to promote its economic development, or explore a new approach based on Afghan characteristics and comparative advantages in order to relieve its dependency on foreign aid, and forge a transition toward stabilization, and eventually, sustainable development.

## 5. BRI as an Alternative to the Traditional Development Aid Model

The BRI, also known as the Silk Road Economic Belt and the 21st Century Maritime Silk Road initiatives, offer opportunities to China and the rest of the world to operate and promote global economic collaborations. It was first proposed by Chinese President Xi Jinping in September 2013. The BRI consists of building a network of railways, highways, oil and gas pipelines, power grids, Internet networks, and aviation routes worldwide. It focuses on building interconnectivity and cooperation among countries, which are alongside the 'Belt' and the 'Road'. The initiative is also an opportunity for China to play a critical role in facilitating regional security and global economic development.

The BRI also aims to assist with the development of on-Road states such as Afghanistan. Specifically, its function is to build the momentum for endogenous economic growth in these states. The traditional development aid model that has operated in Afghanistan over the past decades has failed because it does not emphasize internal dynamism for promoting the sustainable economic development. The traditional development aid model is not aimed at building viable economic activities, basic infrastructure, and bilateral and multilateral agreements that lead to the job creation and economic status improvement among nations, which are closed to the future of the Afghan people and government. Conversely, the BRI facilitates a series of economic intentions that are robustly supported by international communities. It is not only an economic initiative, but also a political proposal that claims to promote cooperation among the on-Road nations, including Afghanistan and its neighboring countries. For example, the Chinese government annually offers higher education programs to Afghan university students. The Afghan students go to Chinese universities for both undergraduate and post graduate courses, which are funded by the Chinese government. This facilitates the potentiality of Afghan development in the future.

In Western Asia, the BRI aims to rebuild Afghanistan as a transit and trade hub. In general, it is different from the traditional development aid model that assists the Afghan economic development in two ways: first, the BRI prioritizes the issue of internal dynamism in stabilizing Afghan society and sustaining its economic development. Second, the BRI recognizes the characteristics of fragility, conflict-affected, and the high unemployment rate of Afghanistan that are long-standing problems caused by the previous national development strategy. Establishing a platform for business collaboration would enhance the economic status of Afghanistan. The BRI also capitalizes on historical issues and integrates disparate reconstruction plans to strengthen Afghanistan's economic development by using the concept of the ancient Silk Road.

*5.1. An Initiative to Facilitate a Regional Transit and Trade Hub*

Historically, Afghanistan has been recognized as the center of the Silk Road. It was treated as the transit hub that bridged the ancient Orient and the Western world. Due to continual conflict, business in Afghanistan has been considerably reduced. Although its geographical advantages provide the potential for independent development in rebuilding Afghanistan as a transit and trade hub, it has not received the attention of either international or local communities. Since 2001, Afghanistan has been consistently building its transportation system and promoting interconnection in West Asia. The BRI has brought structural opportunities to Afghanistan. The Afghan government is voluntary in its cooperation with China. The Chinese government also believes that Afghanistan could play an important role in facilitating a new form of collaboration between China, Central Asia, West Asia, South Asia, and Europe. This provides the opportunity for sustainable development in Afghanistan and other on-Road countries.

In particular, the BRI provides the financial and technical support, human capital, and material resources to establish this transit hub. In 2013, China started to participate in the construction of the highway project between Kabul and Jalal Abad, which is the main corridor for Afghanistan to connect with Pakistan. Rail freight between China and Afghanistan has been set up and trade air-routes have also been established. The accomplishment of these projects has also facilitated an opportunity to form transit and trade agreements with Asian countries, especially China and Pakistan, to allow Afghanistan to participate in their economic activities. The BRI further supports the progressive implementation of International Road Transport (TIR) in Afghanistan. As Janan Mosazai said: Afghanistan is looking forward to peace and prosperity [18]. It should become a trade hub and market rather than the battlefield under the win-win situation with China. The BRI offers opportunities for the future prosperity of Afghanistan.

*5.2. An Initiative to Promote Industrial Upgrades and Resource Corridors*

The Afghan economy is heavily dependent on foreign aid. Although agriculture is the main industry of the Afghan economy and accounts for almost one quarter of its GDP, the amount of food produced is not sufficient to meet demand. Afghanistan has to import food and look for food aid to address continual shortages. As a vast agricultural country, China is experienced in food production. Afghanistan could draw upon the agricultural knowledge of China while China also provides the food aid.

From an industrial perspective, growth is slow and accounts for around 20% of the GDP. Afghan industry is mainly comprised of small and medium-sized enterprises (SMEs), which include light industry and handicrafts. The foundation for sustaining economic growth is fragile and a sound industrial system is lacking. Domestic energy resources are rich, but the utilization rate is very low and barely affects the growth of the GDP in Afghanistan.

The Ghani administration has treated the development of the energy and mineral sectors as a priority in the Afghanistan National Developing Strategy. The Afghan government has successively launched large mineral, oil, and gas development projects, and the development process has accelerated significantly. The BRI also includes a component of energy co-development projects, in particular, cooperation with large mineral and natural gas projects. At present, Afghanistan has cooperated with China on three major projects, including the Aynak Copper Mine project.

The BRI provides unprecedented opportunities to upgrade industry and rebuild the resource corridors in Afghanistan. It also facilitates commercial activities between Afghanistan and its neighboring countries. The main components and efforts of the initiative are to enable policy and regulation in the local society to encourage cross-border business and investment. It would also strengthen the function of the government, which could help to improve their economic status and bolster investment.

### 5.3. An Initiative to Develop the Private Sector

Enterprises are the foundation of the development to a market economy development. The development of business contributes to the sustainable development of the economy, especially SMEs and the private sector. The development of SMEs and the private sector provide continual support for sustaining the growth of trade and job opportunities in Afghanistan. First, the BRI forms a platform for Afghanistan to facilitate cooperation among the SEMs and the private sectors. With the support of the BRI, China and Pakistan have built an information exchange platform in the economic corridor between China and Pakistan. They exchange news, policies, regulations, etc., from both countries in order to share data, reduce the information asymmetry in the field of energy investment, and thus improve the efficiency of investment and cooperation. In the future, Afghanistan could establish an information exchange and sharing platform with potentially collaborative countries under the support of the BRI. It assists with investments in Afghanistan that support the development of Afghan private enterprises, improve local employment, and promote economic transformation.

Second, the BRI provides technical support for the development of Afghanistan's private sector. The Afghan government has established an Integrated Trade and Small-Medium Enterprise Support Facility to provide management expertise to private enterprises [24]. The BRI not only directly brings China's more advanced technology and management experience to Afghanistan through this platform, but also provides a platform to share technologies and experiences among countries to spur the growth of SMEs through the joint Private Advisory Committees and Regional Chambers of Commerce in Central and South Asia.

Finally, the BRI provides financial support for private sector growth in Afghanistan. In 2014, China's foreign direct investment (FDI) in Afghanistan was USD 27.92 million and the total amount of China's FDI in Afghanistan was USD 518 million by the end of 2014 [46]. In the financial year of 2013–2014, besides several major investment projects, there were 49 projects valued at a total of USD 58 million in Afghanistan. In 2014, the Chinese government proposed a four-year 2-billion RMB investment plan for Afghanistan during Ghani's China visit [47].

### 5.4. An Initiative to Develop the Regional Labor Market

Since 2001, a number of industrial parks have been constructed and rebuilt in the cities of Kabul, Nangahar, and Herat. With the increase in bilateral trade and international investment, the demand for workers has also increased, which requires the migration and mobility of skilled workers [46,47]. Previous rules, regulations, and policies on labor migration have been obstacles to improving the productivity and development of the labor market in Afghanistan. The BRI provides empirical, political and administrative assistance in developing policies, judiciaries, and moral terms to process the procedures for legal migration in the region. This improves the legality of the administration and rules in labor migration, and enables orderly labor movements to promote regional cooperation. The perfection of labor migration laws is crucial to reduce poverty, increase job opportunities, improve social stabilization and spread knowledge and skills in Afghanistan and the countries on the Road.

From an administrative perspective, the BRI offers technical and financial support to the Afghan government to introduce an information system for management and analysis in the labor market and also aims to promote employment. This improves the understanding of the labor market in Afghanistan and its neighboring countries. It then facilitates the capacity of policy making to manage and control the labor market [48,49].

The essentiality of the BRI is to encourage improvements in internal dynamism during the economic development process, which increases the confidence and willingness of investors, which leads to job creation. Activities, especially those that facilitate a regional transit and trade hub, development of the private sector, promote industrial upgrades and resource corridors, and promote the status of the regional labor market are critical to enable the production and trade that drive economic growth, social development and long-term regional security.

*5.5. Potential Obstacles to BRI in Afghanistan*

The findings identified above provide a better understanding of the internal dynamism of development for promoting the processes of the BRI. The findings suggest, following the use of the traditional development aid model to assist in the development of Afghan society and economy that additional indicators can provide a more comprehensive understanding of local circumstances. Instead of the traditional development aid model, the BRI can be used to rationally and progressively emphasize the endogenous issues and settings in sustaining the Afghan economy. In particular, the core philosophy of the BRI is to facilitate economic development in accordance with the internal dynamism, and that issues with the local situation are crucial to the generation of successful outcomes. However, some obstacles influence the success of the BRI.

First, security is one of the main obstacles in the implementation process of the BRI. One clear intent of the BRI is to enhance the interconnectivity among the countries that lie on the Road. BRI uses large infrastructure construction projects to promote international cooperation. The Afghan government is also interested in participating in these economic activities. However, it is difficult for China to protect its personnel and assets through the general mechanism of overseas interest protections due to the domestic security situation. This is slowing the implementation of the BRI in Afghanistan. For example, a railway between Kandahar and Jeman (Pakistani) has been promised, and the highway between Kabul and Peshawar has yet to be built because of security concerns. The highway from Kabul to Jalalabad has been slowly built due to the safety issues [50]. Although the BRI has been cautious to proceed, this does not mean that China or the BRI will withdraw from Afghanistan. The Chinese government has made several statements about accelerating the progress of the projects. In order to promote regional security, Afghanistan must sustain economic growth and create job opportunities. On the premise of non-interference in its internal affairs, the Afghan government should engage in political and diplomatic affairs to solve the problems of complex politics and security. This is crucial for Afghanistan to establish a safe political background for large-scale economic investment and trade cooperation in the long term.

Second, communication with the local communities and tribes has also influenced the progress of the BRI in Afghanistan. For instance, the Aynak Copper Mine project has been interrupted several times. Besides national security issues, land acquisition, cultural relics excavation, and other issues are obstacles to the implementation of this project [26]. Therefore, Chinese institutions need to avoid having a negative impact on the local society and appropriately manage local relationships while promoting its role as an investor. The BRI needs to assist the Afghan government in strengthening coordination with its local departments. In particular, the Afghan government in Kabul and the Taliban partly control half of Afghanistan's authority, with positive outcomes at recent peace talks. The BRI should be concerned about its relationship with the authorized Afghan government as well as the Taliban.

## 6. Conclusions

The BRI was founded to build a platform for global economic collaboration and development, but it has been influenced by Chinese over-production capacity and the purpose of international alliance. This has led to an attempt to promote the economic firms of countries that lie on the Belt and Road. This study undertook a detailed comparison between the traditional development aid model and the BRI, which both attempt to stabilize and sustain economic development using Afghanistan as the case study. The analysis here suggests that the traditional development aid model is insufficient to explain the status of Afghan society whereas the BRI is more relevant.

In addressing research question one, this paper found that the world and international communities may have changed their version on fragile states, but the approach to sustain the economic development of Afghanistan and other fragile states did not change. Without sustainable development both in society and the economy, Afghanistan is highly likely to revert to conflict. To answer the second research question, the paper further observed that the traditional development model has proved to

be utterly deficient in Afghanistan because it utilizes a 'one size fits all' approach that has failed to design specific solutions to the unique circumstances of the country; it has also caused Afghanistan to remain disproportionately dependent on foreign aid rather than initiate its internal dynamism to generate economic growth. The answer to research question three is that the BRI prioritizes the economic aspects of development by focusing disproportionately on political and security issues, including building an international cooperation platform and increasing the employment numbers, etc. These are to 'win the hearts and minds' of ordinary people. Finally, the answer to the last research question is that the BRI aims to mobilize indigenous power to promote sustainable development in society and the economy rather than the traditional development aid model that usually leads to the 'Dutch disease'.

In summary, this paper makes four contributions. First, it explores the insufficiencies of the traditional development aid model in stabilizing fragile and conflict-affected states such as Afghanistan. Second, this paper evaluated the BRI as a policy model to facilitate endogenous growth and promote sustainable development. Third, it established analytical and methodological variables that can be used to understand the impacts of the BRI. Finally, this research provides an understanding of fragile and conflict-affected states through the lens of foreign aid and international collaboration. Overall, this paper presents the first comprehensive study of the BRI as an approach that assists with stabilization and sustainable development in Afghanistan. It has important implications for globalization, international development, regional security and international development aid literature, since the BRI aims to remedy and improve upon the insufficiencies in the traditional development aid model.

This research has three significant limitations: first, this study only focused on one conflict-affected state, which was Afghanistan; second, the data collected for this paper were limited and only selected from government sectors; and finally, given the status of the interviewees, the interviews were confidential. Moreover, this study did not collect data from the economic and business sectors. Those weaknesses will be addressed in future studies and will be the focus of future research by the author.

**Author Contributions:** Conceptualization, Y.Z.; Methodology, Y.Z., X.Y. and H.Z.; Validation, Y.Z., X.Y. and H.Z. Formal Analysis, Y.Z., X.Y. and H.Z.; Investigation, Y.Z.; Resources, Y.Z., X.Y. and H.Z.; Data Curation, Y.Z., X.Y. and H.Z.; Writing–Original Draft Preparation, Y.Z., X.Y. and H.Z.; Writing–Review & Editing, Y.Z.; Visualization, Y.Z., X.Y. and H.Z.; Supervision, Y.Z. and H.Z.; Project Administration, Y.Z., X.Y. and H.Z.; Funding Acquisition, Y.Z. and X.Y.

**Funding:** Major project of key research bases of Humanities and Social Sciences of the Ministry of Education in China: "One Belt One Road Initiative and National Strategy among China, Russia and Mongolia" (17JJDGJW006) and Jilin University (451170302045). and Major project of key research bases of Humanities and Social Sciences of the Ministry of Education in China: Chinese strategies on The Korean peninsula (2017JJDGJW005).

**Conflicts of Interest:** The authors declare no conflict of interest.

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
