# Peer review of "Addressing the Insufficiencies of the Traditional Development Aid Model by Utilizing the One Belt, One Road Initiative to Sustain Development in Afghanistan"

_sustainability, doi:10.3390/su11020312_

Round 1

Reviewer 1 Report

Thank you for the opportunity to read the article, the topic is very interesting but I have the following remarks:

1) There is no overview of prior studies, so you must add a literature review showing who dealt with the same topic, who did research projects on that topic and what are the current knowledge in this field. What are the prior studies? How your article will find the gap in our knowledge?  [I am afraid, this is a must!]

2) In final conclusions you must have four elemenets: (i) summary of empirical results, (ii) implications and redcommendation for practice, (iii) research limitations and (iv) suggestion for further studies. You must add more. What are the research laminations of your article? Why we should take your findings with the caution? What are the weakest points of your line of reasoning?  What topics should be undertaken by you and other researchers in the future.  [I am afraid, this is a must!]

3) The article is very descriptive, so it is not scientific. It doesn't use scientific methods such as statistical correlations or regressions. If you use public statistics you show only figures, but you don't correlate these variables and show more advanced picture of our reality (Unfortunately in my opinion this is a typical descriptive article in traditional IR or political science, and not the article in social sciences suitable for IF journals). This paper is written by a political scientists, but by an economists as there are no economic methods used in the article. Please try to enhance the article by applying more sophisticated and more advanced methods (e.g. econometric and mathematical forecasting  of international trade).

4) As you write "The core hypothesis of this paper is to evaluate whether the ‘OBOR’ initiative can replenish 66 the insufficiencies of traditional developing aid model to sustain the development in Afghanistan.". Unfortunately this is NOT a hypothesis, this is an objective and a research qurestion in one. You must reformulate, reshape it.  

5) YOu must retrevie your hypothesis from the literature and you must verify it by using advanced statistical techniques/tests.

6) There is no aim/objective of the paper in the abstract (lines 11-24). The aim in the intorudction (line 42) is different than in the methodology section (line 58).

7)Please provide your research questions in the introduction of your article.

8) Proofreading is a must, e.g. "Author also conducts..." -> "The authors also conduct ...."

To cut a long story short, in its current form, I would think that the paper is more suitable to a political science journal (I mean it is more like an essay than an original reserach article in social sciences including economics, business, not to mention hard science).

Author Response

Dear reviewer:

Thank you for your comments. And I have corrected all the errors by questions as follow:

1) The new literature Reviews part has been done in section 3;

2) A new conclusion including a completed new paragraph have been written in the conclusion in accordance with your comments;

3)Yes indeed, I am a scholar in public policy studies and international relationship. I am really not good at using the econometric and mathematical forecasting model and etc. but the policy analysis. Even though, I try my best to give some sophisticated pictures to make this paper more scientific in the revision. (See line 256, 344 and 438);

4)I have rewrite it as “The core hypothesis of this paper is whether the ‘OBOR’ initiative can facilitate the internal dynamism and then promote the sustainable development in Afghanistan? Hence, the objective of this paper is to evaluate whether the ‘OBOR’ initiative can replenish the insufficiencies of traditional developing aid model to sustain the development in Afghanistan. It is a tendency that the fragile and conflict-affected states look for an approach in facilitating the internal dynamism to develop their economy and society while the foreign aids reduce“;

5)As you know it is hard to a public policy people of using advanced statistical techniques/tests, and I write this paper as public policy and governance paper. I have put more sophisticated pictures to make this paper more scientific in the revision. (See line 256, 344 and 438);

6)The research aim has been written as “The aim of this research is to evaluate whether the ‘OBOR’ initiative can replenish the insufficiencies of traditional developing aid model to sustain the development in Afghanistan”

The research aim in methodology was revised as “The central theoretical aim of this paper was to assess the important issue of the OBOR initiative to replenish the “blind-spot’ of the traditional developing aid model in order to stabilize and sustains the development in Afghanistan.” in accordance with the research aim in introduction;

7)The research question has been added as “The research question of this paper is whether the OBOR initiative is more advanced than the traditional development aid model in sustaining the development in Afghanistan” in the introduction part;

8)The paper has undergone with Specialist proofreading by MDPI experts.

Thank you very much again for your help

Reviewer 2 Report

Authors present the characteristics and developments gained through a programme of public interventions called OBOR, in the context of Afghanistan's economy. This programme is supported by foreign aids by Chinese government and is characterised by an approach quite different from the traditional development aid model.

Said that, I must confess that this article cannot be considered neither as a research article nor a review article, becaduse:

a) there is no theoretical background which is discussed and developed;

b) the reseach method is just mentioned but it is not described at all;

c) there is no evidence about the conclusion authorshave developed in the final part of the text.

Consequently, OBOR results are only presumed but not really proved and documented.

Moreover, English language is not adequate and the text needs a deep revision.

Author Response

Dear reviewer:

Thank you for your comments. And I have corrected all the errors by questions as follow:

1)The new literature Reviews part has been done in section 3;

2)The status of Interviewees have been added as your recommends. This can make this paper more statistical;

3)A new conclusion including a completed new paragraph have been written in the conclusion in accordance with your comments. To be honest, OBOR initiative is ongoing policy production. It is new and functions well in the on-road countries. We have seen some positive outcomes, and that is why this special issue want to conclude some papers on this topic. Moreover, The paper has undergone with Specialist proofreading by MDPI experts,

Thank you very much again for your help

Reviewer 3 Report

This article is well organized.  I expected it would be as well written as it was well structured; I was wrong, for reasons I will present below.  Its road map is clear and meaningful.  It delineates the Afghan economy into four justifiable periods: before 1978, 1979-2001, 2002-2012, after 2013.  It also identifies reasons for the failure of the traditional development aid model, and views OBOR as a response to that failure.  All is well and good—generally

.

The limitations of this work.

1.         Requires extensive (repeat, extensive) editing.  Every page has strong indications of violation of standard grammatical rules of the English language.  Yes, every page.  Some examples from only the first page lf this submission:

·             Line 11 should read make up (two words) as a verb; makeup (solid) as a noun and adjective.

·             On line 14, for the prepositional phrase beginning with “one of the” and ending with “who,” “which,” or “that,” the real subject of the dependent clause is the noun or pronoun following “of.”  Therefore, it should read “have been . . .”  Still awkward nonetheless.  Advice: recast entire sentence.

·             Line 16: The authors also conducted interviews with political elites and policymakers in . . .”  

·              Line 34: recast to read “since September 11, 2001” or simply as “since 9/11.”

·             Line 35: recast to read “flow into Afghanistan as foreign aid, the Afghan people . . .”

2.         It is no longer OBOR; it is BRI.  Update throughout paper.

3.         The four periods of the Afghan economy need tweaking: before 1978, 1978- (NOT 1979), and on and on.

4.         In 3.5, establish a much stronger, more direct argument that economic and social development must be preceded by peace.  The Russians were on the ground for decades, then the Americans.  Peace and tranquility, then development projects can take off.  That was alluded to on line 215: “rebuilding the legitimacy of government.”   Also on 3.5 use concrete examples to substantiate your arguments. 

5.         On lines 276-284, provide concrete examples.  Concrete examples.

6.         Deep-six the use of “lys” throughout paper: First (not Firstly), second (not secondly).

7.         Provide a profile of your interviewees.  How were their views indicated (or reflected) in the findings of this work?  Unclear.

Author Response

Dear reviewer:

Thank you for your comments. And I have corrected all the errors by questions as follow:

1)All comments have been done. And the paper has undergone with Specialist proofreading by MDPI experts;

2)These have been done;

3)Yes, It has been corrected;

4) A new argument has been added as "Failure to establish an approach that promotes the employment, government revenues, and incomes sufficient for Afghans to support their country’s growing population greatly increases the likelihood of an intensified insurgency and internal conflict. Until the economy is divorced from the international aid, the risk of the current conflicts and the turbulent situation of society spilling over into civil war increase successively";

5) An example has been added as "For example, the Chinese government offers the higher education program to Afghan university student annually. The Afghan students go to Chinese universities for both undergraduate and post graduate courses, which are funded by Chinese government. This facilitates the potentiality of Afghan development in the future";

6)All corrections have been done and the paper has undergone with Specialist proofreading by MDPI experts,

7) Their status have been added in the footnotes

Thank you very much again for your help

Reviewer 4 Report

Since the focus is on OBOR, it must be specified that it is the anagram of One Belt and One Road initiative (OBOR). Furthermore, on the subject there is a rich and recent scientific production focused on China that can also be considered to enrich the bibliography.

In the introduction the originality of the work should be better clarified.
In the paragraph methods and materials the methodology used to process the data referred to in the introduction has not been specified. Clearly there is no consistency between the two parts of the work. Therefore, in the paragraph it is necessary to specify in detail who the interviewees are and how the information and the data collected have been treated.
In section 4.5, it starts erroneously speaking of empirical results.
In the concluding paragraph you should better specify the practical implications and the limits of the research.

Author Response

Dear reviewer:

Thank you for your comments. And I have corrected all the errors by questions as follow:

1)It has been verified as BRI in accordance with other reviewers suggested;

2)Some new ideas have been added into the introduction part;

3)The status of interviewees have been added in footnotes

4)The"empirical" has been deleted;

5) A new conclusion including a completed new paragraph have been written in the conclusion in accordance with your comments

Thank you very much again for your help

Round 2

Reviewer 1 Report

Dear Authors,

Thnak you, as I can see huge progrerss in your revised article. In order to make your paper publishable please include the following changes:

1) Introduction ection must be longer, you must expain the novelty of the article in here, as well as the objective of the article must be included in here.

2) I can't accept hypothesis/hypotheses if no staistical/econometric calculations are made. It is against the scientific logic. Please delete hypothesis and ask reserach question (RQ) or a couple of explorative RQ, that would be much better for your article. Please do not forget to mention and answer these RQs in the conclusion section.

3)    Are there any theoretical approaches (public policy theories) which you could use to explain the foundations of OBOR? For now only the concepts and their operationalizations are explained.

Good luck!

Author Response

Dear Reviewer: 

Thank you very much for your help and supports. It is really appreciated. And my changes in line with your kindly comments are as follow:

1.The introduction section has been enriched from 302 words to 727 words. The novelty has been explained here as:

” The findings of this study support a novel policy approach that is expressed in the concept of the BRI. The initiative situates Afghanistan as a regional trade and transit hub on the Belt and Road. The projects constructed through the BRI have significantly increased Afghanistan’s per capita gross domestic product, government revenues, and employment in Afghanistan. Moreover, the study further finds that Afghanistan must relieve its dependence on traditional development aid and facilitate its internal dynamism of economic growth to promote sustainable development. Overall, the novelty of this paper is to present the first comprehensive study of the BRI as a policy production in facilitating the sustainability of conflict-affected states such as Afghanistan”. 

And the objective is included as:

“The objective of this paper is to explore the insufficiencies of the traditional development aid model that have led to its failure in stabilizing the social status and sustaining the economic development of Afghanistan. It further aims to clarify that the BRI can supplement the traditional development aid model. Moreover, it also aims to verify that the BRI is more effective than the traditional development aid model in facilitating internal dynamism for promoting economic development.”

2.The hypotheses have been deleted. And there are four research questions have been placed in the methodology part. 

First, what is the situation and characteristics of the Afghanistan’s economy? 

Second, what are the insufficiencies leading to the failure of the traditional development aid model in Afghanistan? 

Third, how the BRI remedies the deficiencies in the traditional development aid model and sustains the social and economic development in Afghanistan? 

And finally, what is the core difference between the traditional development aid model and the BRI in sustaining the economy of Afghanistan?

Moreover, the answered have been provided in the conclusion part as” In addressing Research question one, this paper found that the world and international communities may have changed their version on fragile states, but the approach to sustain the economic development of Afghanistan and other fragile states did not change. Without a sustainable development both in society and economy, Afghanistan is highly likely to revert to conflict. To answer the second research question, the paper further observed that the traditional development model proved utterly deficient in Afghanistan because it utilized a ‘one size fits all’ approach that failed to design specific solutions to the unique circumstances of the country; it also caused Afghanistan to remain disproportionately dependent on the foreign aid rather than initiate its internal dynamism to generate the economic growth. The answer to research question three is that the BRI prioritizes the economic aspects of development by focusing disproportionately on the political and security issues including build the international cooperative platform and increase the number of employment, etc.. These are to ‘win hearts and minds’ of ordinary people. And finally, the answer to the last research question is that the BRI aims to mobilize the indigenous power to promote the sustainable development in society and economy rather than the traditional development aid model that usually lead the ‘Dutch disease’”. 

Furthermore, MDPI experts have proofread them all.

3.The fragile states theory has been used as the main framework of this paper. 

Finally, Thank you very much indeed for your advises especially in the time of the year. Merry Christmas and Happy New Year. 

Reviewer 2 Report

Although the theoretical background has been marginally improved, sorry to say that major critical points (lack of a research method and especially no evidence at all about the conclusions authors have developed) still remain as they were in the previuos version.

Consequently, OBOR results remain only presumed but not really proved and documented.

English languahe has been also improved. 

Author Response

Dear Reviewer: 

Thank you very much for your help and supports. It is really appreciated. 

According to your comments, I have rewrote the introduction part, enriched the methodology part and provided a more detailed conclusion to answer the research questions and support the research aims. And All of them have been proofread by MDPI experts. 

Finally, Thank you very much indeed for your advises especially in the time of the year. Merry Christmas and Happy New Year.

Reviewer 4 Report

No further changes to the paper are required

Author Response

Dear reviewer:

Thank you very much for your advises. Your help and kindness are highly appreciated by us. If there is no further change to be required about this paper, would you mind to sign the review report?

And May I merry Christmas and Happy New Year to you

Best of Wishes

Round 3

Reviewer 1 Report

Dear Authors,

Thank you for including my comments and good luck!

Happy New Year!

All the best

Reviewer 2 Report

Sorry to say that, but the authors' revision has not significantly improved the quality of the article. Authors do not seem to understand that their major problem is not the use of qualitative analysis, but the lack of a sound methodology, the description of this methodology and the lack of information supporting their statements.

In conclusion, the analysis of BRI seems just an apology for the policy initiative implemented by the Chinese governement, rather than the critical and objective analysis of a new approach in a fragile state. 

These fundamental limitations are still there and have not solved by the second revision.